# Research on the High-Temperature Stability of Twin-Screw Desulphurised Rubber Powder Composite SBS-Modified Asphalt and Its Mixtures

**DOI:** 10.3390/ma18030480

**Published:** 2025-01-21

**Authors:** Dongna Li, Yongning Wang, Jingzhuo Zhao, Fucheng Guo, Bo Li, Tengfei Yao

**Affiliations:** 1School of Mechanical Engineering, Lanzhou Jiaotong University, Lanzhou 730070, China; lidongna@mail.lzjtu.cn (D.L.); fcguo@lzjtu.edu.cn (F.G.); libo@mail.lzjtu.cn (B.L.); tfyao@lzjtu.edu.cn (T.Y.); 2Gansu Provincial Transportation Planning Survey and Design Institute Co., Ltd., Lanzhou 730030, China; 3School of Ecology and Environment, Xinjiang University, Ürümqi 830046, China

**Keywords:** twin-screw desulphurised rubber powder composite SBS-modified asphalt, high-temperature performance, multiple stress repeated creep recovery test, immersion Hamburg rut test, fluorescence microscope

## Abstract

To analyse the differences in the high-temperature performance of twin-screw desulphurised rubber powder/undesulphurised rubber powder composite SBS-modified asphalt and its mixes. This paper analyses the performance differences between desulphurised rubber powder composite SBS-modified asphalt (ACR/SBS), rubber powder composite SBS-modified asphalt (CR/SBS) and SBS-modified asphalt and their mixtures by multi-stress repeated creep recovery (MSCR) and submerged Hamburg rutting tests. In addition, fluorescence microscopy was used to reveal the micro-mechanisms underlying the differences in the high-temperature performance of the three asphalts. The results show that the twin-screw desulphurisation of rubber powder can significantly improve the high-temperature performance and water damage resistance of its composite-modified asphalt and mixture. The rutting depth of ACR/SBS-MA mixes was one-third and one-thirteenth of CR/SBS-MA mixes and SBS-MA mixes, respectively, under the hydrothermal coupling condition at 80 °C. The cross-linking bonds were opened during the twin-screw desulphurisation process to form a stable cross-linking network structure with SBS. The research of this thesis can lay theoretical and technical support for the promotion and application of desulphurised rubber-modified asphalt.

## 1. Introduction

Recycled rubber powder is not easily compatible with asphalt due to its complex internal network cross-linking structure [1,2], which leads to poor high-temperature stability, is not easy to construct and presents other problems, limiting its further popularization and application. To improve the high-temperature performance of rubber asphalt, scholars mainly use chemical desulphurisation, physical desulphurisation and other means of pretreatment of rubber powder. Some scholars pre-treat the gum powder by means of a twin-screw extruder to improve its desulphurisation degree. Enhance the compatibility between rubber powder and asphalt so as to improve the high-temperature stability of rubber-modified asphalt [3,4,5,6]. However, twin-screw desulphurisation can make the internal cross-linking structure of the rubber powder is opened while ensuring that the main bond is not destroyed, and it is considered to be the ideal desulphurisation method. It also makes up for the rubber asphalt’s high-temperature stability being poor, its low dosage and its other problems [7,8].

Regarding the modification mechanism and performance evaluation of asphalt modified by twin-screw desulphurised rubber powder, Si [9], using a twin-screw extrusion method for the preparation of recycled rubber, found that the internal three-dimensional network structure of the powder activated by screw extrusion was partially opened, the cross-linking density was reduced and the mechanical properties of the recycled rubber were optimal at the extrusion temperature of 220 °C. Yazdani [10,11] and others found that the twin-screw extrusion temperatures ranged from 190 °C to 280 °C, and the properties of recycled rubber and modified bitumen were both affected. Due to the late start of the research related to desulphurised rubber powder-modified asphalt, the current scholars mainly use asphalt softening point, dynamic shear rheology (DSR) and other tests to evaluate and analyse the high-temperature stability performance of desulfurization rubber powder-modified asphalt [12,13,14,15,16]. However, there are fewer studies related to the high-temperature performance of rubber powder-modified asphalt mixtures with twin-screw desulphurisation. In particular, there are fewer studies on the high-temperature performance of desulphurised rubber powder composite SBS-modified asphalt mixtures. In addition, the existing microwave/chemical desulphurisation methods for the treatment of rubber powder have a low degree of desulphurisation and are not easy to produce on an industrial scale. Twin-screw extrusion is a thermo-mechanical activation method that breaks the macromolecular chain of rubber powder under high temperature and shear. Twin-screw extrusion method has the characteristics of low pollution, continuous production, etc., and has better application prospects.

For this reason, this paper for the current twin-screw desulphurised powder composite SBS-modified asphalt mixture high-temperature stability performance is not clear, the use of multi-stress repeat creep recovery (MSCR) test and different temperatures of the flooding Hamburg rutting test comparative analysis of its performance with SBS-modified asphalt and ordinary ACR/SBS between the performance differences. This study aims to investigate whether twin-screw-activated rubber powder, when combined with SBS, achieves superior rutting resistance under high-temperature and water-heat coupling conditions.

## 2. Materials and Methods

### 2.1. Raw Materials

Zhenhai 90 base asphalt, 425 μm (40 mesh) waste bias tire rubber powder and 1301 linear SBS modifier, furfural extracted oil and stabilizer were used to prepare SBS-modified asphalt (SBS-MA), CR/SBS-modified asphalt (CR/SBS-MA) and ACR/SBS-modified asphalt (ACR/SBS-MA), respectively. The technical index of the base asphalt and rubber powder are presented in Table 1 and Table 2. Base asphalt and mastic powder from Gansu Provincial Transportation Planning Survey and Design Institute Co., Ltd. (Lanzhou, China).

The preparation process of modified asphalt and twin-screw activated rubber powder is as follows: (1) The activator and the extracted oil are heated and pretreated separately before use; they are placed in the oven at 75 °C for 1 h to reach the Newtonian fluid state. (2) The activator and the extracted oil are mixed thoroughly, and then the resulting mixture is poured into the gum powder and stirred well using a mixer. (3) We left the well-mixed gum powder at room temperature for 24 h to allow it to fully develop. (4) We put the developed rubber powder into the twin-screw extrusion equipment to activate the granulation potential. We activated the rubber granule preparation process, as shown in Figure 1.

Test method for reference JT/T 797-2019 [17], the solubility of the activated rubber powder was tested to be 49.2%.

In this study, SMA-10 mix type was used to mold each type of asphalt mixture and the gradation used for the test is shown in Figure 2. According to the designed mineral proportion batching, 5.7%, 6.0%, 6.3%, 6.6% and 6.9%, a total of five oil–rock ratios were used for the volumetric index test. According to the test results, the relationship curves of gross bulk density, void ratio, stability, saturation, mineral gap ratio and oil–rock ratio were plotted, respectively. Determination of the optimum asphalt content of the gradation was also carried out, and its optimum asphalt content was 6.3%.

### 2.2. Test Methods

(1)Temperature scanning test

The temperature scanning test is conducted according to the method T0628 in the specification JTG E20-2011 [18], and the complex shear modulus G* and phase angle δ of asphalt is obtained.

(2)Multi-stress repeated creep recovery (MSCR) test

After short-term ageing of the three modified bitumens by DSR according to AASHTO T350 [19], the MSCR tests were conducted using the short-term aged specimens at temperatures ranging from 52 °C to 82 °C. Reference [12] performed the calculations for the MSCR test. The average creep recovery rate R and the average irrecoverable elasticity Jnr were calculated for 10 cycles per pair of stress levels according to Equations (1) and (2), where *N* is the creep recovery period corresponding to N is the creep recovery period, corresponding to 21 to 30 for 3.2 kPa.(1)$$R_{3.2}=\frac{\sum_{N}^{10}\varepsilon_{r}\left(3.2,N\right)}{10},N=\left(21,\;30\right)$$(2)$$J_{nr3.2}=\frac{\sum_{N=1}^{10}J_{nr}\left(3.2,N\right)}{10},N=\left(21,\;30\right)$$

The lower the Jnr value and the higher the R value of the asphalt, the stronger its resistance to permanent deformation.

(3)Water immersion Hamburg rutting test

The immersion burger rutting test was carried out using a DWT machine manufactured by IPC Global, Boronia, Australia. The DWT test evaluates the high-temperature performance and water stability properties of three modified asphalt mixtures. The diameter of the test steel wheel is 203.1 mm, the width is 47 mm, the test temperature is 70 °C and 80 °C, the axle load is 1.4 MPa and the termination condition is 40,000 times of rolling of the test steel wheel or the rutting depth reaches 25 mm.

(4)Fluorescence microscope test

Distribution of desulphurised rubber powder and SBS modifier in asphalt binder at microstructural level using fluorescence microscopy. DFM-66C fluorescence microscope of Shanghai Caikang Optical Instrument Co., Ltd. (Shanghai, China) was used to obtain the photos of fluorescence microscope of different modified asphalt. The emission wavelength of the fluorescence microscope was selected from 450 to 490 mm, the exposure time was 1 s, the light intensity was set to 12CD, and the magnification was 100 times. The test process is shown in Figure 3.

## 3. Results

### 3.1. Asphalt Conventional Performance Test Results

Needle penetration, softening point and viscosity tests were used to evaluate the performance differences between SBS-modified asphalt (SBS-MA), rubber powder compounded SBS-modified asphalt (CR/SBS-MA) and desulphurised rubber powder compounded SBS-modified asphalt (ACR/SBS-MA). The results of the tests are shown in Table 3.

In the comparisons of SBS-MA, CR/SBS-MA and ACR/SBS-MA, three kinds of asphalt indicators can be found: the ACR/SBS-MA rather than CR/SBS-MA softening point significantly improved, indicating that the high-temperature performance of the composite modified asphalt prepared after the desulphurisation of the gum powder was significantly improved, and it was higher than the SBS-MA. A comparable 5 °C ductility can be found in the ACR/SBS-MA when the ductility is 39 cm. More specifically, 39 cm is much higher than CR/SBS-MA and slightly higher than SBS-modified asphalt, indicating that the rubber powder desulphurisation after the ductility was significantly improved. In addition, ACR/SBS-MA can still be maintained at 27 cm after ageing, indicating that it has a better ageing resistance. The ACR/SBS-MA separation softening point difference was only 0.5 °C; this is, when compared to CR/SBS-MA, significantly reduced and higher than SBS-modified asphalt, indicating that the composite modified asphalt prepared through the twin-screw desulphurisation of the rubber powder has good storage stability.

### 3.2. Results of Temperature Scanning Test

The complex shear modulus G* and phase angle δ of three asphalt binders, SBS-MA, ACR/SBS-MA and CR/SBS-MA, were tested by DSR at different temperatures (52, 58, 64, 70, 76 and 82 °C) and the results are shown in Figure 4.

From the test results in Figure 4, it can be seen that the composite modulus of all three asphalt binders decreases with the gradual increase in temperature, indicating that the deformation resistance of these three asphalt binders decreases gradually with the increase in temperature. A comparison of the three asphalt binder complex modulus laws can be found; regardless of the temperature, the ACR/SBS-MA complex modulus is always higher than SBS-MA and CR/SBS-MA, and the SBS-MA complex modulus is the smallest, which indicates that the rubber powder added to SBS-MA can improve high-temperature performance. Desulphurative rubber powder’s contribution to the improvement in SBS-MA’s high-temperature performance is more obvious. This is mainly because the rubber powder in the desulphurisation process of C-S and S-S bonds are opened, and SBS forms a more stable spatial network structure so that its high-temperature stability performance is significantly improved.

Differences in the three asphalt binders’ phase angle results can be found. ACR/SBS-MA and CR/SBS-MA showed a gradual increase in δ with the increase in temperature, and SBS-MA showed an increase in temperature δ overall in a stable state; the change is small, indicating that the addition of rubber powder in SBS-modified asphalt can increase its viscosity component. Differences between CR/SBS-MA and ACR/SBS-MA can be found; the ACR/SBS-MA phase angle increase rate is significantly greater than the CR/SBS-MA, indicating that the rubber powder prepared by the desulphurisation of composite modified asphalt viscosity increases, showing more viscous components. The viscous component of the desulphurised rubber powder leads to better compatibility and stability of the powder with asphalt. It shows that in a certain temperature range, the higher the temperature, the better the stability performance of desulphurisation rubber powder-modified asphalt.

On the basis of the previous paper, the rutting coefficient (|G*|/sinδ) is used as a parameter to characterise the ability of the three asphalts to resist permanent deformation at high temperatures. The larger the |G*|/sinδ, the stronger the ability of the asphalt binder to resist deformation, and vice versa. The results of |G*|/sinδ for the three modified asphalts at different temperatures are shown in Figure 5.

The above study shows that the rutting coefficients of the three asphalt binders show a gradual decrease with the gradual increase in temperature. This indicates that the high-temperature rutting resistance of the three asphalt binders gradually decreases with the gradual increase in temperature. Under the same temperature, the rutting coefficients of the three asphalts were analysed, and it was found that the rutting coefficients of ACR/SBS-MA were significantly higher than those of SBS-MA and CR/SBS-MA, which indicated that ACR/SBS-MA had better rutting resistance under the same temperature conditions. The DSR test revealed that the desulphurised rubber powder would significantly improve the high-temperature rutting resistance of the desulphurised rubber powder composite SBS-modified asphalt.

### 3.3. Multi-Stress Repeated Creep Recovery (MSCR) Test Results

The high-temperature performance of the three asphalt binders was evaluated using the recovery rate, R, and the irrecoverable softness, Jnr, tested by the MSCR test. The MSCR test in DSR was carried out with three parallel samples and was very reproducible within 0.5%. The standard deviations of Jnr and recovery were 0.4% and 0.3%, respectively. The smaller the irrecoverable creep softness, the better the high-temperature performance. The larger the creep recovery rate, the better the deformation recovery ability. The high-temperature performance of the three modified asphalt at 3.2 kPa load was tested, and the recovery rate R was the same as that of the MSCR test. The change curves of recovery rate R and unrecoverable soft volume Jnr with the temperature of three modified asphalts under 3.2 kPa loads are shown in Figure 6.

From Figure 6a, it can be seen that the unrecoverable creep flexibility of the three kinds of asphalt binder showed a gradual increase with the gradual increase in temperature, and the rate of increase in SBS-MA was significantly higher than that of CR/SBS-MA and ACR/SBS-MA. At 82 °C, the unrecoverable creep flexibility of three kinds of asphalt binders, namely, SBS-MA, CR/SBS-MA and ACR/SBS-MA, were 9.50, 2.98 and 0.27, respectively. The unrecoverable creep flexibility of SBS-MA and CR/SBS-MA were 45 and 10 times higher than that of ACR/SBS-MA, respectively. It shows that the high-temperature performance of ACR/SBS-MA and CR/SBS-MA is better than that of SBS-MA, and the higher the temperature, the more prominent the high-temperature performance. Figure 6b shows that the creep recovery rate of the three kinds of asphalts shows a gradual decreasing trend with the increase in temperature. CR/SBS-MA and ACR/SBS-MA show a stable and then rapidly decreasing trend, while SBS-MA shows a rapid decline and then tends to stabilise. It shows that SBS-MA basically loses its high-temperature performance when the temperature is higher than 76 °C and no longer changes with the increase in temperature.

### 3.4. Rutting Test Results of Immersion Burger

In the analysis of three kinds of asphalt binders’ high-temperature stability performance on the basis of the three kinds of asphalt rutting, using immersion burger rutting test three kinds of asphalt mixtures in 70 °C water bath rutting depth change rule, the results are shown in Figure 7.

From Figure 7, it can be seen that the rutting depth of the three asphalt mixtures under the 70 °C water bath condition increases gradually with the number of axial load (1.4 MPa) actions. The number of rutting points at the damage for the three asphalt mixtures were 17,900, 30,000 and 36,200 times. This indicates that SBS-MA and ACR/SBS-MA mixtures have better rutting damage resistance than CR/SBS-MA mixtures. In addition, comparing the rutting depths at 17,900 axle loads, it can be found that the ACR/SBS-MA mix has the largest rutting depth, the CR/SBS-MA mix has the second largest rutting depth, and the ACR/SBS-MA mix has the smallest rutting depth. This indicates that the three mix specimens have the best resistance to deformation before damage, the ACR/SBS-MA mix has the best resistance to deformation, and the ACR/SBS-MA mix has the best resistance to deformation before damage. The modified mix has the worst deformation resistance, which is a good indication that the ACR/SBS-MA mix has better high-temperature stability performance at 70 °C hydrothermal coupling conditions. This also shows that the ACR/SBS-MA blend has good high-temperature stability and water stability at 70 °C hydrothermal coupling.

To further evaluate the high-temperature stability performance of the three asphalt mixtures, 1.4 MPa and a 80 °C water bath were used to simulate their deformation capacity under high-temperature and heavy-loading conditions, and the results are shown in Figure 8.

From Figure 8a, it can be seen that the rutting depths of the three asphalt mixtures in the water bath at 80 °C gradually increase with the increase in the number of axle loads (1.4 MPa). The SBS-MA mixtures are the first to be damaged, followed by the CR/SBS-MA mixtures, and the ACR/SBS-MA mixtures are the last to be damaged. This further indicates that the ACR/SBS-MA mixture has better resistance to water damage. Moreover, a 80 °C water bath gives better resistance to water damage than a 70 °C water bath for the CR/SBS-MA mixture or the SBS-MA mixture, which is mainly due to the fact that the softening point of SBS-MA is only 74 °C, which leads to its softening and early damage in a 80 °C water bath. As can be seen from Figure 8b, the increase in rutting depth with the increasing number of loadings is significantly higher for SBS-modified asphalt mixtures than for CR/SBS-MA and ACR/SBS-MA. The increase in rutting depth with the increasing number of loadings is the smallest for ACR/SBS-MA mixtures, which is one-third and one-thirteenth of that of the increase in rutting depth for CR/SBS-MA and SBS-modified asphalt mixtures, respectively. This shows that the resistance to high-temperature deformation of ACR/SBS-MA mixes increases significantly with the increase in the number of loadings.

### 3.5. Analysis of Fluorescence Microscopy Results

To explain the reason for the improved high-temperature stability of ACR/SBS-modified asphalt, fluorescence microscopy images of three types of asphalt, CR/SBS-modified asphalt, SBS-modified asphalt and ACR/SBS-modified asphalt, were captured using fluorescence microscopy, and the results are shown in Figure 9.

We carried out five parallel tests on each sample by fluorescence microscopy and found that it had good reproducibility, and finally, we selected a representative one for comparative analysis. As can be seen from the fluorescence microscope image in Figure 9, the SBS modifier and rubber powder particles in CR/SBS-modified asphalt are agglomerated and unevenly distributed. The SBS modifier and rubber powder particles in SBS-modified asphalt and ACR/SBS-modified asphalt are more evenly dispersed. This is mainly due to the action of high-temperature shear, which means the twin-screw activation powder internal cross-linking bond is destroyed. The rubber molecular chain is also constantly broken into small-chain segments uniformly dispersed in the matrix asphalt system, and then it forms a stable cross-linking network structure, showing good high-temperature performance.

## 4. Conclusions

(1)Rubber powder by twin-screw desulphurisation will significantly improve the high-temperature rutting resistance of its composite-modified asphalt. SBS-modified asphalt will significantly improve at a temperature of more than 76 °C with the basic loss of high-temperature performance, and this will cease with the increase in temperature. The rubber powder composite-modified asphalt by twin-screw desulphurisation at 82 °C still has a good creep recovery rate.(2)In 70 °C hydrothermal coupling conditions, twin-screw desulfurisation of rubber powder composite SBS-modified asphalt mixture of high-temperature stability and water stability significantly improved.(3)Under the condition of hydrothermal coupling at 80 °C, the rutting depth of ACR/SBS-MA mixtures has the smallest increase rate with the number of loads, which is one-third and one-thirteenth of CR/SBS-MA mixtures and SBS-modified asphalt mixtures, respectively.(4)The desulphurised rubber powder and SBS modifier in twin-screw desulphurised rubber powder composite SBS-modified asphalt form a stable cross-linking network structure in the asphalt matrix, which in turn improves its high-temperature stability performance and water stability performance.(5)The research results of this project can save 10% for construction and maintenance during the whole life cycle and extend the service life of the pavement by 2–3 years. The asphalt dosage is recommended to be determined in combination with the actual gradation of the project.(6)In the next step, we would like to test the durability and ageing performance of FGD composite modified asphalt mixtures under freeze–thaw cycles as well as the monitoring of the engineering practice, with a view to providing technical support and engineering practice for its promotion and application.

## Figures and Tables

**Figure 1 materials-18-00480-f001:**
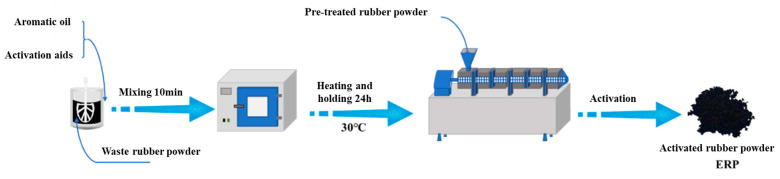
Twin-screw-activated rubber powder process.

**Figure 2 materials-18-00480-f002:**
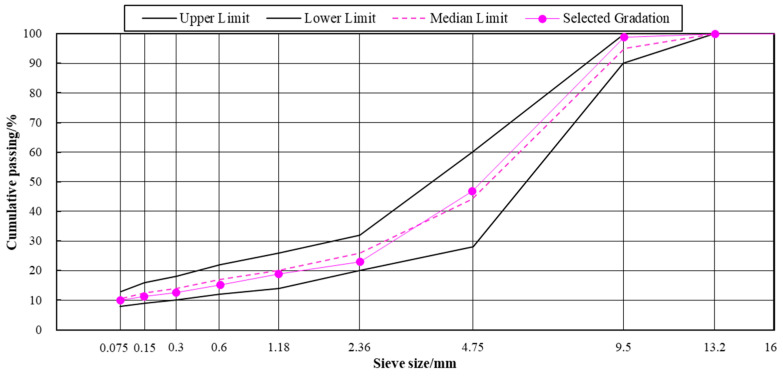
Grading curve.

**Figure 3 materials-18-00480-f003:**
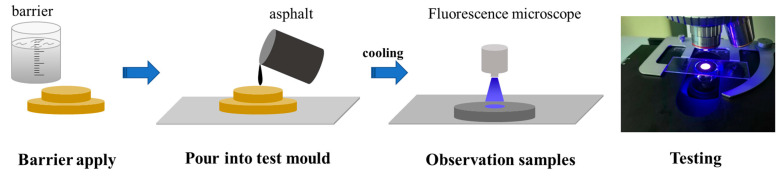
Fluorescence microscopy testing process.

**Figure 4 materials-18-00480-f004:**
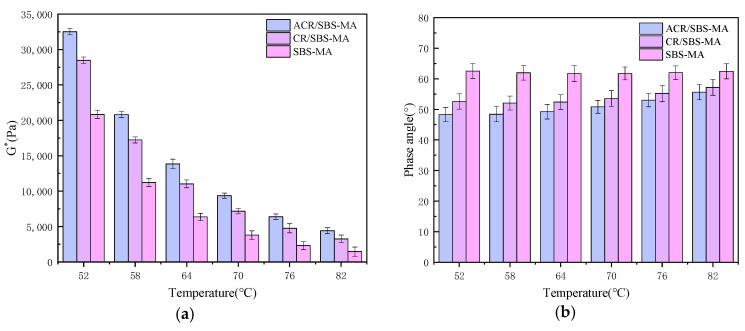
G* and δ of each modified asphalt at different temperatures. (**a**) G*; (**b**) δ.

**Figure 5 materials-18-00480-f005:**
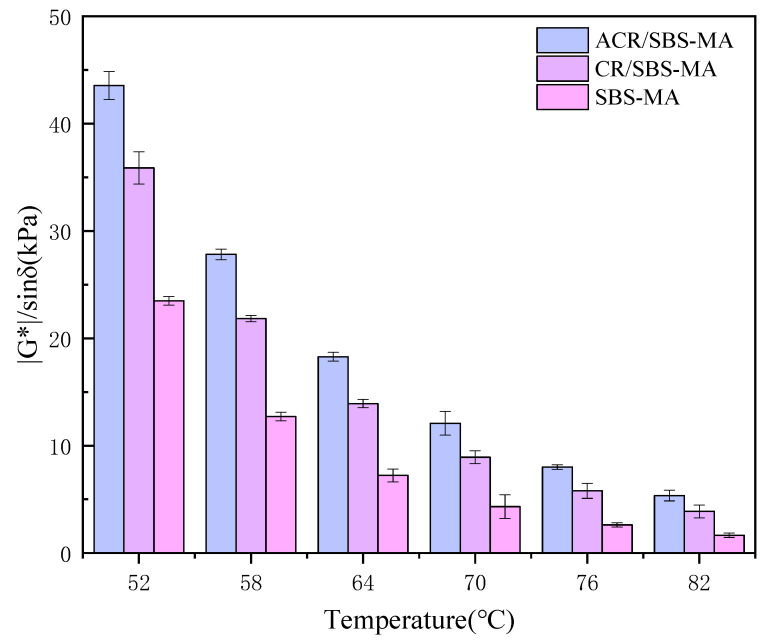
Rutting factors (|G*|/sinδ) of modified asphalt at different temperatures.

**Figure 6 materials-18-00480-f006:**
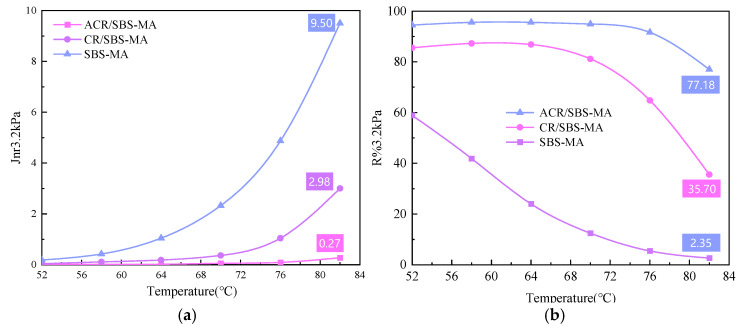
Temperature-dependent curves of irreparable flexibility Jnr and recovery rate R. (**a**) Jnr; (**b**) R.

**Figure 7 materials-18-00480-f007:**
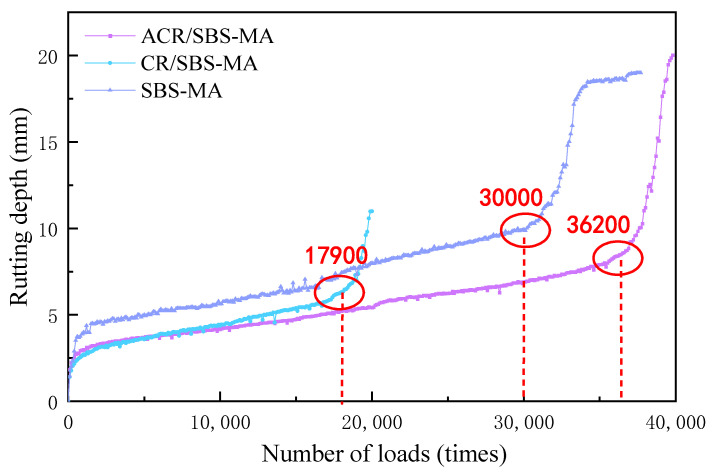
Curve of changes in rut depth with a number of axle loads (70 °C, 1.4 MPa).

**Figure 8 materials-18-00480-f008:**
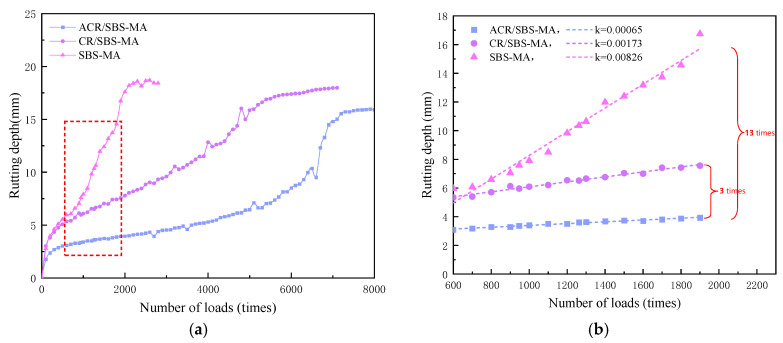
Curve of changes in rut depth with number of axle loads (80 °C, 1.4 MPa). (**a**) Rutting depth; (**b**) rate of change of rutting depth. Figure (**b**) is an enlarged view of the red box.

**Figure 9 materials-18-00480-f009:**
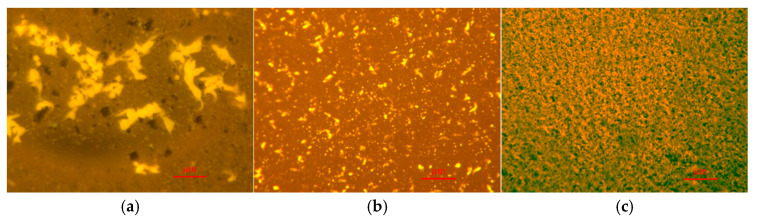
Fluorescence microscope test results. (**a**) CR/SBS-MA; (**b**) SBS-MA; (**c**) ACR/SBS-MA.

**Table 1 materials-18-00480-t001:** Technical index test results of Zhenhai 90 # base asphalt.

Technical Indicators		Unit	Test Result	Required Value	Test Method
Penetration (25 °C, 100 g, 5 s)		0.1 mm	68.7	60~80	T0604
Softening point		°C	46.8	≥46	T0606
Ductility (10 °C)		cm	41.7	≥20	T0605
Ductility (15 °C)		cm	>150	≥100	
Density		g/cm^3^	1.032	/	T0603
60 °C power viscosity		Pa∙s	191	≥180	T0620
RTFOF (163 °C, 85 min)	Mass loss	%	0.12	≤±0.8	T0610
	Penetration ratio	%	68	≥65	T0604
	Ductility (10 °C, 5 cm/min)	cm	6.2	≥6	T0605

**Table 2 materials-18-00480-t002:** Technical index of rubber powder.

Technical Index	Test Result	Required Value
Relative density/(g/cm^3^)	1.15	1.1~1.2
Water content/%	0.52	<1
Iron content/%	0.01	<0.01
Fiber content/%	0.07	<0.5
Ash content/%	7.23	≤8
Acetone extractives/%	6.24	6~16
Carbon black content/%	30	≥28
Rubber hydrocarbon content/%	52	42~65

**Table 3 materials-18-00480-t003:** Technical indexes of modified asphalts.

Index		SBS-MA	CR/SBS-MA	ACR/SBS-MA
SBS content/%		3.0	3.0	3.0
Gum powder content (mass ratio)/%		0	20	20
Penetration (25 °C, 100 g, 5 s)/(0.1 mm)		70	68	73
Penetration index		0.21	3.30	2.20
Ductility (5 °C, 5 cm/min)/cm		30	21	39
Softening point/°C		72	83	91
48 h Softening point difference/°C		1.2	6.9	0.5
After ageing	Mass loss/%	−0.08	0.10	−0.30
	Penetration ratio/%	80	83	84
	Ductility (5 °C, 5 cm/min)/cm	26	14	27

## Data Availability

The original contributions presented in this study are included in the article. Further inquiries can be directed to the corresponding authors.

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
