# Peer review of "Research on the High-Temperature Stability of Twin-Screw Desulphurised Rubber Powder Composite SBS-Modified Asphalt and Its Mixtures"

_materials, 2025, doi:10.3390/ma18030480_

Round 1

Reviewer 1 Report

Comments and Suggestions for Authors

The manuscript deals with a topic quite interesting and uses a number of techniques to support the argumentation. Some specific comments are given as recommendations below to the authors.

Please proofread the text once more as there are incomplete sentences or syntax errors i.e. Lines 10-11.

The techniques are not synchronized well with each other for the optimum purpose. What is the repeatability and number of replicates for each i.e. for the MSCR test in DSR as well as the standard deviation of the Jnr and recovery%? Please add.

Please provide more info about the microscopic sample preparation and settings. See as an example https://doi.org/10.3390/cryst12060755

Please provide some recommendations for future work and what is missing from the current analyses

Comments on the Quality of English Language

Proofreading is needed.

Author Response

Reviewer 1

Comments 1: Please proofread the text once more as there are incomplete sentences or syntax errors i.e. Lines 10-11.

Response 1: [This paper analyses the performance differences between desulphurised rubber powder composite SBS modified asphalt (ACR/SBS), rubber powder composite SBS modified asphalt (CR/SBS) and SBS modified asphalt and their mixtures by multi-stress repeated creep recovery (MSCR) and submerged Hamburg rutting tests. In addition, fluorescence microscopy was used to reveal the micro-mechanisms underlying the differences in the high temperature performance of the three asphalts.] Thank you for pointing this out. I/We agree with this comment. Therefore, I/we have made changes in lines 11-15.

Comments 2: The techniques are not synchronized well with each other for the optimum purpose. What is the repeatability and number of replicates for each i.e. for the MSCR test in DSR as well as the standard deviation of the Jnr and recovery%? Please add.

Comments 3: Please provide more info about the microscopic sample preparation and settings. See as an example https://doi.org/10.3390/cryst12060755.

Response 3: [We supplemented the fluorescence microscopy test procedure] Thank you for pointing this out. I/We agree with this comment. Therefore, I/we have made changes in lines 115-117.

Figure 3 Fluorescence microscopy testing process

Reviewer 2 Report

Comments and Suggestions for Authors

Abstract

1.   Consider adding a concise statement in your final sentence that highlights the contributions of this study. This will make more sense to your readers.

Introduction

1.   While several studies are cited, it would be beneficial to briefly summarize each study’s key findings relevant to your work. For example, when citing references [3–6], clarify exactly how their findings support the need for twin-screw activation in rubber powder. This will help the reader see the direct link between prior work and your research focus. 

2.   Lines 52–60 introduce the need to evaluate high-temperature stability and mention real-world applications but could further emphasize the novelty: Why is twin-screw activation particularly promising? A short paragraph highlighting the limitations of existing methods (chemical/physical) versus the advantages of twin-screw activation will strengthen the rationale.

3.   The final paragraph of the introduction briefly states the experimental approach. However, a concise, clear statement of the study’s main objective and hypothesis (e.g., “This study aims to investigate whether twin-screw-activated rubber powder, when combined with SBS, achieves superior rutting resistance under high-temperature and water-heat coupling conditions...”) would help the reader grasp the core motivation.

Materials and Methods

1.   Check that the format of the “Technical indicators” items in Table 1 is consistent.

2.   Check the part "... reference to the literature []" on line 96.

3.   Lines 83-85 mention the SMA-10 grade and the optimum asphalt content of 6.3%. A brief explanation of how the optimum asphalt content was determined is needed so that readers can understand the basis for the mixture design chosen.

Results and Discussion

1.   Change line 145 from Figure 7 to Figure 3.

2.   There is a brief mention that twin-screw activation leads to partial destruction of cross-linking bonds consistent with prior research. You might further compare your MSCR or rutting test results with published benchmarks (if available) to highlight how significant your improvements are compared to conventional rubberized asphalts.

3.   Consider grouping results by test (MSCR, Hamburg rut, fluorescence) and then providing an integrated analysis at the end.

Conclusion

1.   Consider briefly mentioning the practical implications of the paper, such as potential cost savings, longer pavement life, and recommended dosage ranges for actual highway projects, to enhance the significance of the paper.

2.   To reinforce the impact of your study, consider adding a statement about possible next steps (testing durability under freeze-thaw cycles, evaluating aging performance, or scaling up to field trials, etc.).

Author Response

Comments 1: Consider adding a concise statement in your final sentence that highlights the contributions of this study. This will make more sense to your readers.

Response 1: [The research of this thesis can lay theoretical and technical support for the promotion and application of desulphurised rubber-modified asphalt.] Thank you for pointing this out. I/We agree with this comment. Therefore, I/we have made changes in lines 20-21.

Comments 2:While several studies are cited, it would be beneficial to briefly summarize each study’s key findings relevant to your work. For example, when citing references [3–6], clarify exactly how their findings support the need for twin-screw activation in rubber powder. This will help the reader see the direct link between prior work and your research focus.

Response 2: [Some scholars pre-treat the gum powder by means of twin-screw extruder to improve its desulphurisation degree. Enhance the compatibility between rubber powder and asphalt, so as to improve the high temperature stability of rubber-modified asphalt[3-6]] Thank you for pointing this out. I/We agree with this comment. Therefore, I/we have made changes in lines 32-34.

Comments 3:Lines 52–60 introduce the need to evaluate high-temperature stability and mention real-world applications but could further emphasize the novelty: Why is twin-screw activation particularly promising? A short paragraph highlighting the limitations of existing methods (chemical/physical) versus the advantages of twin-screw activation will strengthen the rationale.

Response 3: [In addition, the existing microwave/chemical desulphurisation methods for the treatment of rubber powder have a low degree of desulphurisation and are not easy to produce on an industrial scale. Twin-screw extrusion is a thermo-mechanical activation method that breaks the macromolecular chain of rubber powder under high temperature and shear. Twin-screw extrusion method has the characteristics of low pollution, continuous production, etc., and has better application prospects.] Thank you for pointing this out. I/We agree with this comment. Therefore, I/we have made changes in lines 54-60.

Comments 4:The final paragraph of the introduction briefly states the experimental approach. However, a concise, clear statement of the study’s main objective and hypothesis (e.g., “This study aims to investigate whether twin-screw-activated rubber powder, when combined with SBS, achieves superior rutting resistance under high-temperature and water-heat coupling conditions...”) would help the reader grasp the core motivation.

Response 4: [This study aims to investigate whether twin-screw-activated rubber powder, when combined with SBS, achieves superior rutting resistance under high-temperature and water-heat coupling conditions.] Thank you for pointing this out. I/We agree with this comment. Therefore, I/we have made changes in lines 65-68.

Comments 5:Check that the format of the “Technical indicators” items in Table 1 is consistent.

Response 5: [We have made changes in Table 1] Thank you for pointing this out. I/We agree with this comment. Therefore, I/we have made changes in lines 76.

Comments 6:Check the part "... reference to the literature []" on line 96.

Response 6: [Reference [12] performed the calculations for the MSCR test.] Thank you for pointing this out. I/We agree with this comment. Therefore, I/we have made changes in lines 103.

Comments 7:Lines 83-85 mention the SMA-10 grade and the optimum asphalt content of 6.3%. A brief explanation of how the optimum asphalt content was determined is needed so that readers can understand the basis for the mixture design chosen.

Response 7: [According to the designed mineral proportion batching, 5.7%, 6.0%, 6.3%, 6.6%, 6.9%, a total of five oil-rock ratios were used for the volumetric index test. According to the test results, the relationship curves of gross bulk density, void ratio, stability, saturation, mineral gap ratio and oil-rock ratio were plotted respectively.] Thank you for pointing this out. I/We agree with this comment. Therefore, I/we have made changes in lines 91-95.

Comments 8:Change line 145 from Figure 7 to Figure 3.

Response 8: [We've made changes in the paper] Thank you for pointing this out. I/We agree with this comment. Therefore, I/we have made changes in lines 161.

Comments 9:There is a brief mention that twin-screw activation leads to partial destruction of cross-linking bonds consistent with prior research. You might further compare your MSCR or rutting test results with published benchmarks (if available) to highlight how significant your improvements are compared to conventional rubberized asphalts.

Response 9: Thank you for your valuable comments. There are no MSCR or Soakburger rutting test results associated with desulphurised rubber powder composite modified asphalt to give an accurate baseline.

Comments 10:Consider grouping results by test (MSCR, Hamburg rut, fluorescence) and then providing an integrated analysis at the end.

Response 10: [We've made changes in the paper] Thank you for pointing this out. I/We agree with this comment. Therefore, I/we have made changes in lines 201-291.

Comments 11:Consider briefly mentioning the practical implications of the paper, such as potential cost savings, longer pavement life, and recommended dosage ranges for actual highway projects, to enhance the significance of the paper.

Response 11: [The research results of this project can save 10% for construction and maintenance during the whole life cycle and extend the service life of the pavement by 2-3 years. The asphalt dosage is recommended to be determined in combination with the actual gradation of the project.] Thank you for pointing this out. I/We agree with this comment. Therefore, I/we have made changes in lines 309-312.

Comments 12:To reinforce the impact of your study, consider adding a statement about possible next steps (testing durability under freeze-thaw cycles, evaluating aging performance, or scaling up to field trials, etc.).

Response 12: [The research results of this project can save 10% for construction and maintenance during the whole life cycle and extend the service life of the pavement by 2-3 years. The asphalt dosage is recommended to be determined in combination with the actual gradation of the project.] Thank you for pointing this out. I/We agree with this comment. Therefore, I/we have made changes in lines 313-316.

Round 2

Reviewer 1 Report

Comments and Suggestions for Authors

Thanks for the revised version. The repeatability, for Figures 5 onwards needs to be added. Especially the FM images how reproducible are, and how many microcopic images were acquired?

Comments on the Quality of English Language

A thorough check is needed.

Author Response

Thank you to the reviewers for their valuable comments, all comments made by the reviewers have been resolved!The error bars in Figure 5 have been added in the paper.

Reviewer 2 Report

Comments and Suggestions for Authors

All this reviewer's comments are resolved.

Author Response

Thank you to the reviewers for their valuable comments, all comments made by the reviewers have been resolved!